# Growth monitoring and promotion program services utilization patterns between home-based and facility-based delivery methods: A comparative analysis

Muttaquina Hossain[1,2]*, Md. Ahshanul Haque[1], Abu Syed Golam Faruque[1], Tahmeed Ahmed[1,3], Ulla Ashorn[2], Per Ashorn[2,4]

1 Nutrition Research Division, icddr,b, Dhaka, Bangladesh, 2 Center for Child, Adolescent and Maternal Health Research, Faculty of Medicine and Health Technology, Tampere University, Tampere, Finland, 3 James P Grant School of Public Health, BRAC University, Dhaka, Bangladesh, 4 Department of Paediatrics, Tampere University Hospital, Tampere, Finland

* muttaquina@icddrb.org, muttaquina@gmail.com

## Abstract

### Background

In Bangladesh, utilization of government health facility-based growth monitoring and promotion (GMP) services is notably low, prompting non-governmental organizations (NGOs) to collaborate with the government to enhance GMP service utilization through home-based delivery. Despite this, there is limited information available on home-based GMP service utilization. This study aimed to investigate the utilization of GMP services between home-based and facility-based programs and identifying key factors and barriers to caregiver engagement with GMP services in rural Bangladesh.

### Methods

A descriptive mixed-method study was conducted across six sub-districts from August to December 2019. Three sub-districts with home-based GMP services provided by NGOs were compared with three neighboring sub-districts offering government facility-based GMP services. A total of 3038 randomly selected mothers and children under one year old were included in the quantitative part of the study. Quantitative surveys include information on household socio-demographic information, GMP service components, knowledge, utilization, barriers, and qualitative approaches were used for data collection on caregivers and service providers perspectives on GMP services. Descriptive statistics were conducted for sociodemographic characteristics, caregivers' knowledge, perception and barriers to utilization of GMP services. Student's t-tests and chi-square tests were used to compare quantitative and qualitative variables between both GMP arms. Risk ratios (RR) with 95% confidence intervals were calculated to compare GMP knowledge. Simple logistic

**Data availability statement:** All relevant data are within the paper and its Supporting Information files. The supporting information file contains data and dataset, which is anonymized and contains no identifiable information.

**Funding:** This research study was funded by USAID under grant no. AID-388-A-17-00006. The funder had no role in study design, data collection and analysis, decision to publish, or preparation of the manuscript.

**Competing interests:** The authors have declared that no competing interests exist.

regression identified GMP service use levels and related barriers. Multiple logistic regression was employed to determine statistically significant associations between GMP utilization and independent variables such as caregivers who heard about GMP or GMP cards, were members of an NGO, and lacked interest in GMP services at p-value <0.05 and adjusted risk ratio (ARR) values. Thematic analysis of qualitative data was performed. Results were triangulated across sources.

## Results

Children's average age was 9.8 months, with a 1:1 male-to-female ratio in both groups (home-based vs. facility-based: 51.9% vs. 50.0%). Home-based GMP services exhibited higher utilization rates, with more children receiving weight and length measurements and caregivers receiving counseling than facility-based services (40% vs. 0% utilization, respectively). Caregivers' utilization of GMP services in home-based areas was positively influenced by their knowledge of GMP or GMP cards (Adjusted risk ratio, ARR: 37.4) and their involvement with an NGO, association, or health program (ARR: 1.3). Caregivers in home-based GMP areas relied on NGO staff for service delivery, while those in facility-based areas reported no outreach from government health workers and lacked access to GMP cards due to supply issues. Across both areas, low awareness of GMP services and the absence of incentives contributed to limited utilization.

## Conclusion

GMP service utilization remains low in rural Mymensingh district of Bangladesh. Home-based GMP service utilization was 40% but none of the caregivers utilized facility-based GMP services. Higher utilization in home-based areas was linked to caregiver awareness, access to GMP cards, and NGO involvement, while key barriers included lack of government outreach, supply gaps, and absence of incentives. To improve GMP coverage, government programs should enhance community level outreach, ensure consistent supply of growth cards, and consider integrating small incentives to motivate caregivers.

## Introduction

Growth Monitoring and Promotion (GMP) is a key part of child health programs that involve regularly measuring a child's growth to detect early signs of growth faltering and support timely interventions. The GMP program is designed to include regular check-ups for child growth and happens every three months at health facilities. In an ideal GMP program, health workers are advised to counsel mothers and caregivers on their children's nutrition, measure the weight, length, or mid-upper arm circumference (MUAC) of all under-2-year-old children, and plot the results on a growth monitoring chart in the GMP card. If a health worker identifies a child as acutely ill or not growing well during GMP, he/she should refer the child to the nearest medical/

treatment center [1]. This approach aims to foster communication and interaction with caregivers, promoting optimal child development [2]. UNICEF's introduction of the promotion component in the mid-1980s strengthened GMP, making it a key part of nutrition programs for managing child malnutrition in low- and middle-income countries [3].

The effectiveness of GMP programs depends on several factors, including program coverage, caregiver interaction with healthcare providers, health worker performance, and facility readiness [4]. Other important factors include community engagement, suitability of the delivery platform, and the broader sociocultural context [2–4]; these factors stresses the need for optimal resource mobilization, such as adequate funding, consistent supply of GMP tools (e.g., growth charts), training and supervision of health workers, and support for community-based outreach efforts [5–7]. In low and middle income countries (LMICs), health facilities primarily implement GMP, so research on GMP programs has mostly focused on identifying deficiencies in facility readiness and supply-side limitations [8–10]. These investigations often neglect the crucial assessment of acceptability and feasibility from the perspectives of caregivers and communities, along with their sociocultural contexts. As a result, the effectiveness of facility-based GMP programs in LMICs in improving caregiver utilization and reducing child undernutrition remains uncertain [1,11,12].

To reduce the gaps in feasibility and utilization of facility-based GMP services, certain LMICs in Africa and Asia have adopted community and home-based platforms for GMP service delivery [1]. Their approaches involve decentralization of health services from facility to bring services closer to the community, making it easier for caregivers to access GMP support without having to travel long distances [10,13–15]. Home-based and community-based models often involve local health workers who provide growth monitoring, counseling, and education directly in caregivers' homes or nearby locations [14]. These models also integrate community health workers to offer personalized support and follow-ups, ensuring continuity of care. The effectiveness of these decentralized approaches hinges upon factors such as the communication and counseling skills of health workers, the motivation and willingness of caregivers to engage with the service, and the accessibility of appropriate referrals and interventions [16]. However, despite the promise of these alternatives, most home/community-based GMP services remain isolated and lack linkage with the broader health system, posing challenges to sustainability and scalability [1,2]. Consequently, the outcomes of home-based GMP services exhibit variability across diverse contexts and regions [13,14].

The Government of Bangladesh provides GMP services through Community Clinics (CCs), where trained staff are expected to monitor all children under five during both illnesses visits and routine check-ups [17]. However, limited active supervision at CCs remains a concern. In contrast, during the study period in Mymensingh, a single NGO partnered with the government to strengthen GMP utilization by delivering home-based services in selected sub-districts. The NGO used a child tracking system, distributed GMP cards to caregivers, and developed counseling tools to support home visits and improve caregiver engagement. The primary objectives are twofold: to assess disparities in GMP service utilization between home-based and facility-based programs, and to identify the key barriers hindering caregiver engagement in each context.

In the ongoing debate surrounding the most effective delivery platform for GMP services in LMICs with limited resources, this study aimed to delve into caregiver GMP service utilization across both home-based and facility-based settings in rural parts of Mymensingh district of Bangladesh. Qualitative approaches are applied to enhance the quantitative survey findings. The results will provide valuable insights into service utilization and the factors influencing caregiver participation, helping researchers, program developers, and policymakers make evidence-based decisions to improve GMP service use and healthcare outcomes in LMICs.

## Materials and methods

### Study design, period, and site

A community-based cross-sectional study employing a mixed-method approach was conducted in six rural sub-districts of Bangladesh's Mymensingh district from August to December 2019. This study is a sub-study of a larger study that aimed to evaluate the effectiveness of GMP programs in improving child nutritional status [18].

We selected the Mymensingh district based on the following criteria: (A) time of implementation of the program matching with the study timeline; (B) staff and logistics in CCs supported by NGO; (C) adequate number of CCs in subdistricts to satisfy the sample size [18]; (D) comparativeness to other Bangladeshi districts with its predominantly rural, agrarian population, along with comparable literacy rates and demographic patterns [18]; and (E) containing health facility-based platforms, notably community clinics (CC) where the Government of Bangladesh (GoB) offers GMP services. These CCs cater to the health, family planning, and nutrition needs of 6,000 or more households [17].

Overall, six rural sub-districts Bangladesh's Mymensingh district were selected. Three neighboring sub-districts were chosen where only the government's facility-based GMP services were available, without any NGO involvement. In addition, to understand the home-based and facility-based GMP service delivery models, three sub-districts were selected where a non-governmental organization (NGO) supported the government's facility-based GMP services with home-based initiatives. In Mymensingh, an NGO collaborated with the government to enhance the GMP program in select sub-districts including the ones selected for this study. The NGO provided home-based GMP services through its staff, using a tracking system for child monitoring. To boost GMP utilization, it developed counseling tools and ensured a consistent supply of GMP cards to caregivers, following standard GMP components outlined in the introduction section of this study [1]. Concurrently, three neighboring sub-districts were chosen where only the government's facility-based GMP services were available, without any NGO involvement.

## Selection of study sites

The home and facility-based areas were matched based on some socio-demographic characteristics to avoid selection bias [18]. Initially, study-specific general information was collected on each CC and its catchment area through discussions with three groups of residents: (1) Community health care providers (CHCPs), (2) mothers/caregivers of under-two children, and (3) members of CC-associated community group (CG). Discussions focused on: (1) GMP service availability, (2) availability of other nutrition services at the CCs for women and children, (3) components of GMP provided by CHCP, and (4) sociodemographic status of people living in the CC catchment area. One point was given to a response if it contains more than or equal to 90% response rate (positive or negative), and less than 90% response got zero, totaling to a maximum of 5 points for a given CC. All the points were then summed to generate a combined score for each CC. Majority of the home-based area CCs (more than 90%) scored 3 out of 5. In the next stage, due to the similarities of scores, 30 home-based area CCs were selected by computer-generated simple random sampling from the total 72 CCs in the NGO-supported subdistricts. Similar scores were generated for each facility-based area CC. The 30 facility-based area CCs were then individually matched with the 30 home-based area CCs based on the closest scores. Thus, the home-based to facility-based CC ratio was 1:1.

We compared caregivers' knowledge of GMP service, levels of GMP service utilization, and barriers to GMP service utilization in both types of settings.

## Selection and recruitment of study participants for the quantitative part of the study

For the quantitative part of the study, all data of participants of the initial, larger study (3038 mother-child pairs) were included [18]. We used proportionate sampling based on the size and population of the sub-district meaning larger study samples were selected from bigger sub-district. Participants were identified through door-to-door screening, and those who met the eligibility criteria were included in the sampling frame. From this frame, participants were randomly selected for the study.

The study included children aged 6–23 months and their mothers/caregivers in CC catchment areas. Eligibility criteria encompassed children free from known acute/chronic illnesses, severe stunting, wasting, or underweight due to ethical obligation as the study would not provide treatment or management, not enrolled in similar programs, and whose caregivers provided consent. The youngest child in each household was designated as the index child, with twins excluded to

prevent potential information biases. In case of participant absence, the next eligible household was approached during data collection.

## Facility-based GMP service delivery

The GoB has implemented a GMP program in its CCs since 2011. Each CC in the current study was meant to have three healthcare staff responsible for providing GMP to every child under five years of age, whether they visited for illness or routine check-ups [17]. The GoB had developed a National Guideline for GMP implementation. It was responsible for training the CC staff and providing the necessary equipment and supplies. Despite consistency in GMP components as described above [1], there has been a concern about the need for more active supervision at the CCs [19].

## Home-based GMP service delivery

During the study period in Mymensingh, a single NGO partnered with the government to support the GMP program in certain sub-districts. The NGO provided extra support by engaging its staff for home-based GMP services. They offered GMP sessions at home for children using a tracking system. The NGO aimed to improve GMP utilization by creating counseling tools and ensuring a steady supply of GMP cards to caregivers at home. The GMP components were aligned with the previously outlined methodology [1].

## Study measures

**Outcome variable: GMP service utilization.** Operational definition: Caregivers' GMP service utilization was defined as whether they received GMP services at home or went to the facility for the service even once during the child's lifetime during the survey.

Exposure variables: Caregiver's knowledge, perception, barriers to access and utilization of GMP services, levels of GMP service utilization, predictors of GMP service utilization. The caregivers' barriers to access and utilization of GMP services were explored using a semi-structured survey questionnaire and during qualitative interviews. The survey questionnaire included multiple response options.

Covariates that were identified as potential confounders included in the models: caregiver awareness of GMP or GMP card, NGO membership, and caregiver's lack of demand for GMP service.

## Data processing and analysis

**Sample size and sampling technique for the quantitative part of the study.** All the data of the participants enrolled in the larger study (3038 mother-child pairs) were included in the quantitative analysis. The detailed sampling methods are published elsewhere [19]. We used proportionate sampling based on the size and population of the sub-district. We excluded children during household screening which is about 16% (n = 291) in home-based area and 7% (n = 122) in facility-based GMP areas who did not meet eligibility criteria.

**Quantitative data collection.** Fourteen data collectors collected quantitative data from participants, using a pretested semi-structured questionnaire (S2 File). The semi-structured questionnaire was developed based on a review of relevant scientific literature and existing tools [20]. Its content validity was reviewed by a panel of nutrition and public health experts in Bangladesh. The original tool was prepared in English, translated into Bangla, and then back-translated to English by an independent translator to ensure accuracy and consistency. It was pretested in a comparable setting to assess clarity, cultural appropriateness, and flow, and subsequently refined before final use. The survey questionnaire covered topics including household socio-demographics, water sanitation and hygiene practices, knowledge and perception of GMP services, utilization of GMP services, perceived barriers in GMP service utilization, and child anthropometry.

Data collectors directly observed households' drinking water sources and toilets. Improved toilets were defined as flush toilets, ventilated improved pit latrines, traditional pit latrines with a slab, or composting toilets; improved drinking water

sources were classified as water from tubewells [21]. The data collectors utilized Bangladesh government-approved GMP cards and asked caregivers about their knowledge of the cards, showing growth charts for interpretation. Caregivers' responses were recorded in close ended questionnaires. If caregivers had not utilized GMP services, they were asked for reasons via a multiple-option choice questionnaire.

Child anthropometry followed WHO standard operating procedures [21], with staff initially trained by certified gold standards at icddr,b. Measurements were taken using a Seca Infantometer for length and a Seca 727 Baby Scale for weight, ensuring daily precision through calibration and repeated readings when necessary. All measuring tools were placed on flat surfaces, and data collectors noted the readings when participants became steady on the scale. One data collector read the measurement, and another recorded it on the form.

**Statistical analysis for the quantitative part of the study.** The completed questionnaires were securely transported to icddr,b and received by data management assistants and a statistician for entry and analysis using MS Access and STATA software version 15, respectively. The study statisticians and investigators decoded caregivers' responses on GMP growth card questionnaires and determined "correct" responses, e.g., if a caregiver mentioned that the "red color zone" in the growth chart means the child is severely malnourished, it was considered "correct" by the investigators. The study team assessed caregivers' responses to GMP growth card questionnaires for accuracy. Child growth parameters were calculated based on WHO standards [22]. A wealth index was created using a composite score based on household assets to categorize socioeconomic status [22]. The assets list used for this index was derived from the Bangladesh Demographic and Health Survey questionnaire [21]. Households were grouped into three categories: rich, middle, and poor, reflecting relative differences in asset ownership.

Data were checked for normality using Skewness/Kurtosis tests, histogram and quantile-normal (Q-Q) plots (qnorm) (S1 File). Outliers were managed according to standard statistical/data analysis procedures. Categorical variables were summarized using proportions and frequencies, while normally distributed continuous variables were summarized using means and standard deviations. Student's t-tests compared counts with a normal distribution, and the chi-square distribution test compared qualitative variables for both home-based and facility-based GMP arms. The differences in caregivers' GMP knowledge between home-based and facility-based GMP proportions were reported using risk ratios (RR) with 95% confidence intervals (95% CI) without adjustments. Simple logistic regression models were used to determine caregivers' level of GMP service utilization and barriers to GMP service utilization.

Predictors of caregivers' GMP utilization were explored using logistic regression models only for the home-based group due to the absence of GMP service utilization among facility-based caregivers. The study examined factors influencing caregiver GMP service utilization versus non-utilization. All independent variables significantly associated with GMP utilization (dependent variable and primary outcome) in bivariate models and biologically plausible variables were included in the final multivariable models. The multivariable models were adjusted for relevant covariates: caregiver awareness of GMP or GMP card, NGO membership, and caregiver's lack of demand for GMP service. Estimated associations were described as adjusted risk ratios (ARR) with 95% confidence intervals (CIs). Results with p-values <0.05 were considered statistically significant.

**Sample size and sampling technique for the qualitative part of the study.** Only a few participants (n = 36) were interviewed for the qualitative part. Sixteen qualitative in-depth interviews (IDIs) were conducted with community health care providers (CHCPs) and 20 with mothers of enrolled children. In addition, daylong indirect observations were conducted at 23 CCs. The qualitative participants were selected purposively from the program areas to supplement the quantitative findings, focusing on gathering in-depth insights into factors influencing GMP service utilization. This approach was chosen to ensure a diverse range of perspectives, particularly those not captured through the quantitative surveys, such as caregivers' personal experiences, beliefs, and barriers to service utilization. All of the participants were interviewed after having written informed consent with a clear explanation of the objective and purpose of the interview. None of the participants refused to participate. The number of qualitative interviews was determined through an iterative

                                                                                                 

process, where data from each round of interviews were analyzed to identify emerging themes. Interviews continued until information saturation was reached—that is, when no new insights were gained, indicating that key themes had been thoroughly explored.

**Qualitative data collection.** Qualitative approaches were applied to enhance the quantitative survey findings. The theoretical framework guiding this study is the Health Belief Model (HBM), which emphasizes individuals' perceptions of health risks, the benefits of preventive actions, and the barriers to adopting these actions. In the context of this study, the HBM helped to understand caregivers' motivations and decision-making regarding GMP services. It informed the development of research questions related to perceived severity, susceptibility, benefits, and barriers to GMP service utilization. This framework also guided the identification of key themes during the qualitative analysis, focusing on the barrier's caregivers face and the factors that influence their engagement with GMP services. By using the HBM, the study aimed to provide a structured understanding of caregivers' behaviors and attitudes toward health service utilization in rural Bangladesh.

The study employed one moderator and one interviewer proficient in conducting qualitative interviews and analysis. The qualitative interviews were carried out by trained moderators and interviewers whose native language was Bangla. The research team consisted of two experts with advanced degrees in public health and nutrition, each having over 10 years of experience in qualitative research, particularly in health systems and nutrition in Bangladesh. These experts have led numerous studies on maternal and child health, health service utilization, and nutrition interventions in low- and middle-income countries. Additionally, the team included two trained coders who were proficient in qualitative data analysis, having received specialized training in coding techniques, thematic analysis, and ensuring consistency in data management. All team members were trained in ethical research practices, culturally sensitive interviewing, and local community engagement to ensure the research was both contextually appropriate and scientifically rigorous. Interviews were conducted in Bangla and later translated into English. The Bangla version of the questionnaire was validated before the interview through translation and back-translation [19]. Open-ended questionnaires and guidelines were used for qualitative interviews, along with checklists for CC observations (S4 File). The interviews with caregivers were conducted at their households with a duration of 30–45 minutes. The interview with service providers were conducted at their duty stations, for 20–30 minutes. The interviews were conducted at convenient times of the participants with proper privacy and absence of non-participants. Repeated interviews were conducted to gather additional information or clarify details from previous interviews with participants. These tools included questions on caregivers' GMP knowledge, perception, and practices, as well as CC-related GMP service information. Two qualitative researchers (Ishrat Jahan and Ziaul Islam) employed a thematic analysis for all qualitative information.

**Analysis of the qualitative part of the study.** Qualitative analysis involved thematic descriptions, analysis, and respondent quotations. Transcribed interviews were coded based on predetermined themes (a priori), subthemes, and emerging issues. The coding process began with thorough transcription and review of the data to familiarize them with the content. A preliminary coding scheme was developed from emerging patterns and concepts. These codes were manually applied to the transcripts, with similar codes grouped into categories. Themes were derived through an iterative process of constant comparison, refining and redefining codes as the analysis progressed. The final themes were shaped by both the research questions and the data. Team discussions and member checks were used to validate the consistency and relevance of the themes. Trustworthiness was ensured through four principles: credibility, transferability, conformability, and dependability [23]. To ensure credibility, prolonged engagement and member checks were used, along with data triangulation to confirm findings. Transferability was enhanced by providing detailed descriptions of the research context, participants, and methods, allowing for comparison to similar settings. Conformability was maintained through an audit trail and peer debriefing to ensure findings were grounded in the data. Dependability was ensured by thoroughly documenting the research process, allowing for replication and consistency over time. Intercoder or synchronic reliability was utilized to gauge the agreement between two independent coders of the data. An agreement was assessed during

the analysis when the researchers independently coded the same interviews. There were no discrepancies observed among the coders. Triangulation was conducted between methods and participants, validating information by triangulating data from different approaches.

### Ethical considerations

The Institutional Review Board (IRB) of icddr,b approved the study (PR-17123, version 1.02, dated 06 March 2018). Written informed consent with full disclosure about the study was taken from the participants before all interviews. The privacy of the respondents, their anonymity, and the confidentiality of data/information were strictly maintained. Only the principal investigators of this study had access to information on personal identification and other sensitive information that the respondents provided.

## Results

### Quantitative survey findings

1.1. Sociodemographic characteristics of the study participants

A total of 3038 mother-child dyads were enrolled, evenly split between home-based and facility-based service areas (Fig 1).
   Children's average age was 9.8 months, with a 1:1 male-to-female ratio in both groups (home-based vs. facility-based: 51.9% vs. 50.0%). Children's length-for-age Z-score (LAZ) and weight-for-age Z score (WAZ) was comparable among the groups. Children from home-based GMP area had significantly lower weight-for-length/height Z-score (WHZ) compared to facility-based area children (mean WHZ: −0.48 versus −0.4, p = 0.04). Mother's age (home-based vs. facility-based: 26.1 vs 25 years) and father's years of education (home-based vs. facility-based: 7.4 vs 6.9 years) were significantly higher in home-based GMP area compared to facility-based GMP area (p = 0.01 and 0.002 respectively) (Table 1). Approximately 98% of household members in both areas identified as Muslim, with 94% and 92% having improved toilets in home-based and facility-based GMP service areas, respectively (p = 0.016) (Table 1). More (44%) caregivers in home-based GMP areas were from poor families compared to facility-based GMP areas (38%) (p = 0.002) (Table 1). The socio-demographic characteristics of the sub-districts in 2 GMP areas were mostly comparable (S1 File).

1.2. Caregiver's knowledge and perception of GMP service

Caregivers from home-based GMP areas had higher GMP knowledge levels compared to those in facility-based areas. In home-based GMP areas, a higher proportion of caregivers (67%) had heard about GMP or GMP cards compared to facility-based GMP areas (0.4%) (RR: 169) (Table 2) (Fig 2). However, only 17% of caregivers in home-based GMP areas correctly understood the purpose of the growth chart and explained the meaning of different colors compared to less than 1% in facility-based GMP areas (RR: 127.5 and 122.5, and 31, 492, respectively). A total of 2,791 (91.8%) out of 3,038 responses from mothers were incorrect; Among them home-based GMP area, n = 1,274 (83.8%) and facility-based GMP area, n = 1,517 (99.8%) responses were incorrect. More than half of the caregivers (51%) in home-based GMP service areas mentioned GMP service as "beneficial", while none from facility-based regions did. The meaning of the word "beneficial" ("*valo/upokari*" in Bangla) was not explained to the respondent, but it was left for them to judge.

1.3. Caregiver's level of GMP service utilization

GMP service utilization was significantly higher in home-based areas compared to facility-based areas. In home-based GMP service areas, 61% of caregivers had received GMP cards (Table 3) (Fig 2), while none had done so in facility-based GMP areas. Significantly more children underwent weight measurement, length measurement, and their caregivers received growth monitoring-specific counseling in home-based GMP service compared to facility-based GMP service (81% vs. 21%; 3.4% vs. 0.6%; and 55% vs. 05%, RR: 4, 5.7 and 9.5, respectively). However, only 30 out of 1519

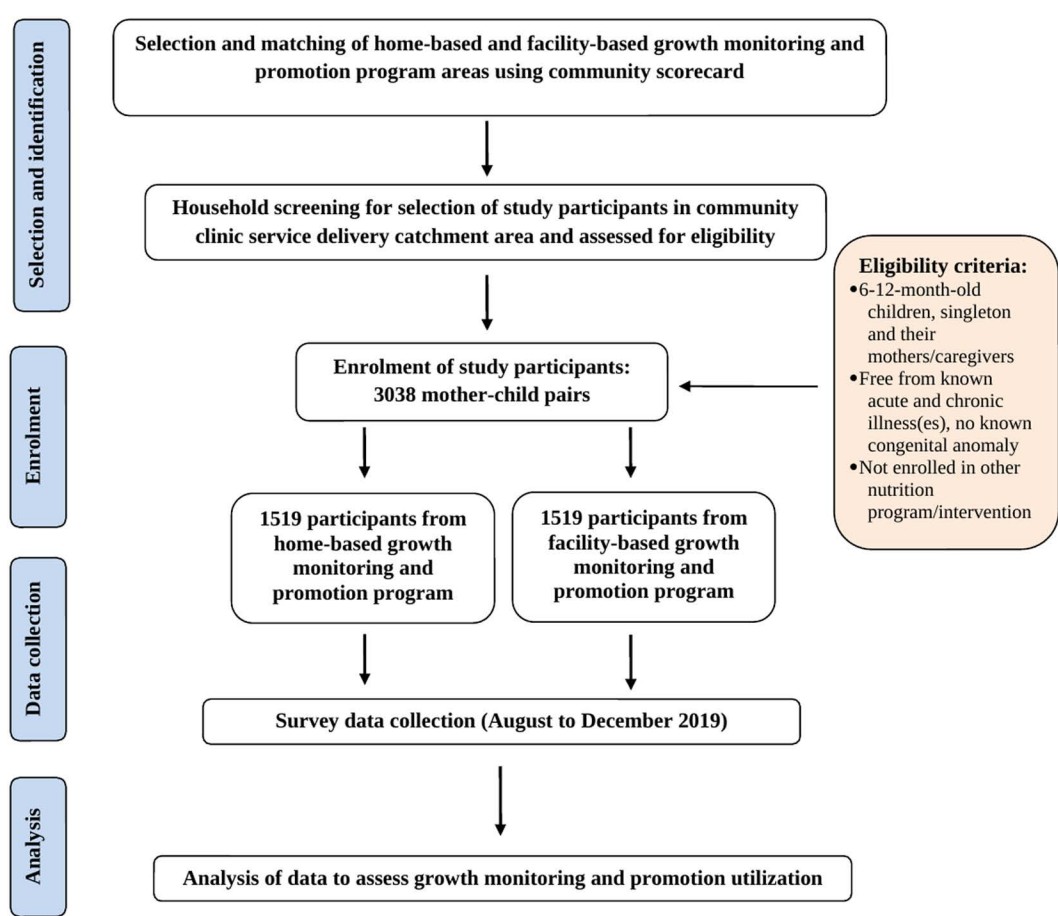

**Fig 1. Participant flow chart.**

children and their caregivers received all three major components of GMP services (child weight, length measurement, and GM-specific counseling) in a single GMP visit at home-based GMP services (Table 3). Children and their caregivers in facility-based areas never received all three components together. Overall, 40% of caregivers from home-based GMP service areas utilized GMP service compared to 0% in facility-based GMP service (Table 3) (Fig 2).

1.4. Barriers to caregivers' access and utilization of GMP services

1.4.1. Quantitative survey findings

Four major barriers were identified as reasons for caregivers' underutilization of GMP services. Among these, caregivers' lack of knowledge about the availability of GMP services at the nearest health facility was a significant barrier in both areas (Table 4). Caregivers from facility-based GMP areas had significantly less knowledge about GMP service availability compared to those in home-based service areas (52% vs. 61%, respectively; RR: 0.9). Additionally, caregivers' demand for GMP service was significantly lower in facility-based areas compared to home-based GMP areas (55% vs. 44%, respectively; RR: 0.7). More caregivers from home-based GMP service areas reported closure of health facilities/absence of service providers (1%) and being preoccupied with household work (4%) as significant barriers to accessing and utilizing GMP services, compared to facility-based GMP service areas (0.2% and 0.3%, respectively) (Table 4).

**Table 1.** Caregiver's sociodemographic characteristics across home-based (N = 1,519) and facility-based (N = 1,519) GMPs.

| Characteristic | Home-based GMP | Facility-based GMP | p-value |
|---|---|---|---|
| Child age in months, mean (SD) | 9.9 (1.7) | 9.7 (1.8) | 0.065 |
| Child's weight-for-age Z-score (WAZ), mean (SD) | −1.1 (1.1) | −1.1 (1.1) | 0.555 |
| Child's length-for-age Z-score (LAZ), mean (SD) | −1.3 (1.1) | −1.4 (1.1) | 0.190 |
| Child's weight-for-length Z-score (WLZ), mean (SD) | −0.48 (0.03) | −0.40 (0.03) | 0.043 |
| Proportion of boys, n (%) | 760 (50.0) | 789 (51.9) | 0.293 |
| Mother's age in years, mean (SD) | 26.1 (5.8) | 25.6 (5.9) | 0.011 |
| Mother's education in years, mean (SD) | 6.1 (3.6) | 6.1 (3.6) | 0.717 |
| Mother's occupation, n (%) | | | |
| Housewife/homemakers | 1,474 (97.0) | 1,487 (97.9) | 0.133 |
| Fathers' education in years, mean (SD) | 7.4 (4.5) | 6.9 (3.4) | 0.002 |
| Religion, n (%) | | | |
| Islam | 1,494 (98.4) | 1,478 (97.3) | 0.046 |
| Hinduism/Buddhism/Christian | 25 (1.6) | 41 (2.7) | |
| Improved toilet facility, n (%) | 1,402 (92.3) | 1,435 (94.5) | 0.016 |
| Improved source of drinking water, n (%) | 1,514 (99.7) | 1,515 (99.7) | 0.739 |
| Wealth index, n (%) | | | |
| Poor | 667 (44.0) | 577 (38.0) | 0.002 |
| Middle | 430 (28.3) | 461 (30.3) | |
| Rich | 420 (27.7) | 481 (31.7) | |

Growth monitoring and promotion, GMP.

**Table 2.** Caregiver's knowledge of GMP service in facility and home-based GMP program.

| Indicators | Home-based GMP N = 1,519 | Facility-based GMP N = 1,519 | RR (95% CI) | p-value |
|---|---|---|---|---|
| Caregivers heard about GMP or GMP card | 1,014 (66.8) | 6 (0.4) | 169 (76, 375.9) | <0.001 |
| Caregivers know the purpose of the growth chart | 255 (16.8) | 2 (0.1) | 127.5 (31.8, 511.6) | <0.001 |
| Caregivers can correctly explain the colors in the GMP card growth chart | 245 (16.1) | 02 (0.13) | 122.5 (30.5, 491.6) | <0.001 |
| Caregivers mentioned GMP service as "beneficial" | 777 (51.2) | 0 (0) | – | – |

Growth monitoring and promotion, GMP; confidence interval, CI; Risk Ratio, RR.

### 1.4.2. Qualitative findings

#### 1.4.2a. *Caregiver's were unaware of the GMP service and its availability*

To compliment the quantitative survey, we conducted qualitative interviews with caregivers of under-2 children, government and CC facility providers on GMP service delivery and utilization. In-depth interviews (IDIs) with mothers/caregivers provided insight into the obstacles to accessing GMP services in both settings. The average age of caregivers were 25 years, had completed 7 years of education, all of them were housewives and had an average monthly family income of 122 US dollars, which is similar to rural Bangladesh context. Mothers/caregivers in the home-based GMP area mainly relied on NGO staff for GMP service delivery at home, bypassing visits to CCs for GMP services. In contrast, mothers in facility-based GMP areas reported that government community health workers never visited their communities or homes to inform them about GMP services at CCs or the nearest health facilities. Caregivers from both home-based and

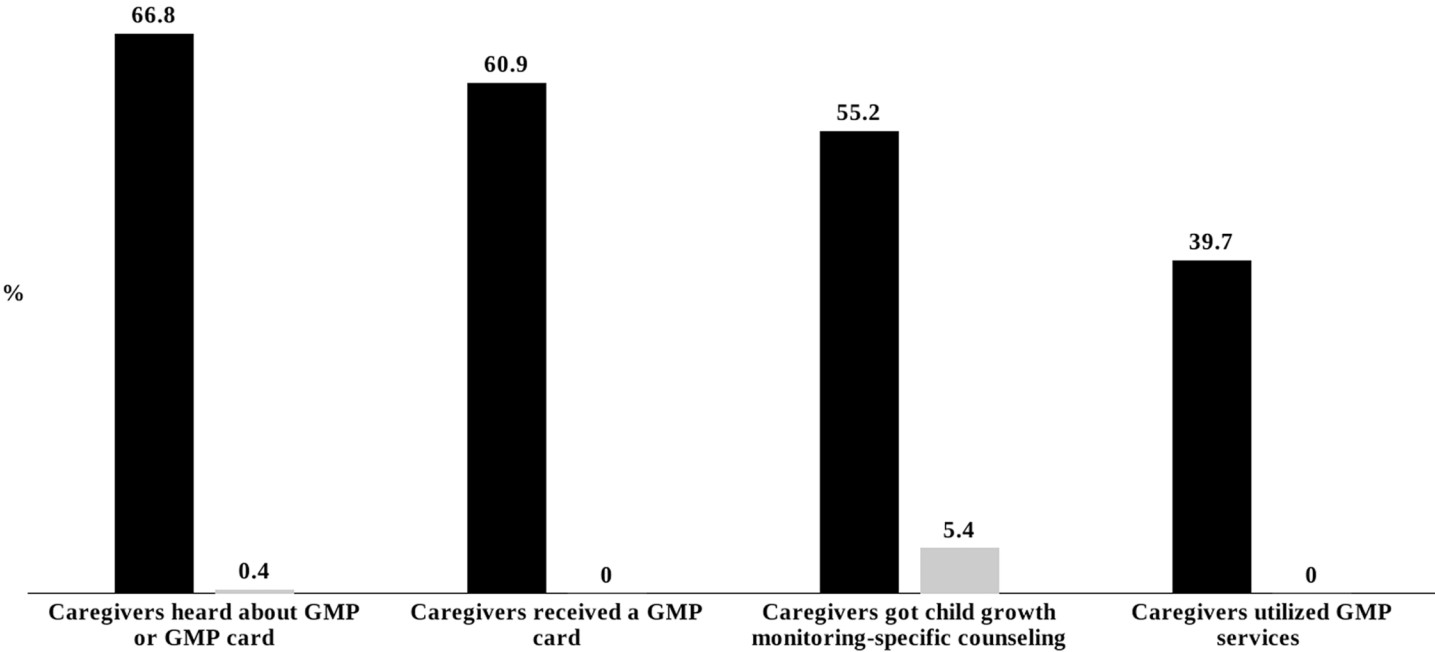

**Fig 2. Caregivers' knowledge and level of GMP service utilization.**

**Table 3. Levels of GMP service utilization among individuals in facility (N = 1,519) and home-based (N = 1,519) GMP programs (N = 1,519).**

| Indicators | Home-based GMP N (%) | Facility-based GMP N (%) | RR (95% CI) | p-value |
|---|---|---|---|---|
| Caregivers received a GMP card | 926 (60.9) | 0 (0) | – | – |
| The child was weighed at least once | 1231 (81.0) | 311 (20.5) | 4.0 (3.6, 4.4) | <0.001 |
| The child length was measured at least once | 51 (3.36) | 09 (0.59) | 5.7 (2.8, 11.5) | <0.001 |
| Caregivers got child growth monitoring-specific counseling | 837 (55.2) | 88 (5.4) | 9.5 (7.7, 11.7) | <0.001 |
| The child received all major components of GMP (length, weight, and growth monitoring-specific counseling) altogether | 30 (1.97) | 0 (0) | – | – |
| Caregivers utilized GMP services | 603 (39.7) | 0 (0) | – | – |

Growth monitoring and promotion, GMP; confidence interval, CI; Risk Ratio, RR.

facility-based GMP areas expressed a general lack of awareness about the availability of GMP services. One caregiver from the facility-based area shared:

"*I didn't know where I can get the service (GMP), no one told me that it (GMP service) is given at community clinic.*" (Caregiver, Facility-Based Area)

This sentiment was echoed by 3 more caregivers from the facility-based area. Caregivers from home-based areas were also unaware of the availability of GMP in community clinics:

**Table 4. Barriers to GMP service utilization among individuals in the facility (N = 1,519) and home-based (N = 1,519) GMP<sup>a</sup> programs.**

| Indicators | Home-based GMP N (%) | Facility-based GMP N (%) | RR (95% CI) | p-value |
|---|---|---|---|---|
| The caregiver was preoccupied with household chores and did not go for the GMP service | 56 (3.7) | 5 (0.3) | 11.2 (4.5, 27.9) | <0.001 |
| The health facility was closed/no healthcare provider at the facility | 14 (0.9) | 3 (0.2) | 4.6 (1.3, 16.2) | 0.015 |
| Caregivers' lack of interest in GMP services at an individual level | 668 (44.0) | 834 (54.9) | 0.7 (0.6, 0.9) | 0.006 |
| Lack of knowledge of GMP service availability at the nearest health facility | 798 (52.5) | 919 (60.5) | 0.9 (0.8, 0.9) | <0.001 |

Growth monitoring and promotion, GMP; confidence interval, CI; Risk Ratio, RR.

<sup>a</sup>Multiple responses by caregivers.

"*The NGO workers visited my home to provide the service, but they didn't mention that GMP is also offered at the community clinic.*" (Caregiver, Home-Based Area)

However, not all caregivers shared this view. A small number of caregivers in facility-based areas mentioned that they had received information about GMP from a local NGO community health worker, illustrating some variation in awareness levels. One caregiver noted,

"*The health worker told me about GMP services during her visit to my home last month*" (Caregiver, Facility-Based Area)

Additionally, caregivers across both areas commonly expressed a preference for services that offered immediate benefits, such as vaccine, food or cash (3 out of 7 caregivers). As one caregiver in a home-based area stated,

"*……rural people want instant benefit, for example- they will go for the vaccination, as they know that it cures the disease. But people don't give importance to measuring (child) length and weight, as it has no instant benefit.*" (Caregiver, Home-Based Area)

This preference was a common theme, though not universal. While most caregivers were unaware of GMP services or preferred services with incentives, there were also outliers who demonstrated awareness and prioritized health over material incentives. A few caregivers indicated they prioritized the health benefits over incentives, saying,

"I go to the clinic because I care about my child's health, not for any money or food" (Caregiver, Facility-Based Area).

1.4.2b. *Lack of GMP card supply in facility-based GMP areas*

Caregivers from facility-based GMP areas never received GMP cards because of lack of supply of GMP cards in CCs from the government. Interviews with government stakeholders revealed the lack of allocation of government funds to print and supply GMP cards to the CCs during the study period. The NGO area caregivers got GMP cards as the NGO printed out and distributed government approved GMP cards using their own funds.

1.5. Predictors of caregivers' GMP service utilization

Utilization of GMP services was positively associated with caregivers' knowledge of GMP or GMP card (Adjusted risk ratio, ARR: 37.4, p=<0.001) and membership of an NGO/association/health program (ARR: 1.3, p = 0.001) (Table 5). A key barrier to GMP service utilization was caregivers' low interest in the service (ARR: 0.7, p = 0.001).

**Table 5. Predictors of GMP service utilization among caregivers at the home-based GMP programs.**

| Characteristic[a] | Characteristic category | RR (95% CI) | p-value | ARR (95% CI) | p-value |
|---|---|---|---|---|---|
| Caregiver having heard about GMP or GMP card | No[1] | – | – | – | – |
| | Yes | 42.4 (20.3, 88.6) | <0.001 | 37.4 (17.8, 78.5) | <0.001 |
| Caregiver being a member of an association/NGO/health program | No[1] | – | – | – | – |
| | Yes | 2.6 (2.1, 3.2) | <0.001 | 1.3 (1.1, 1.5) | 0.001 |
| Caregiver's low interest in GMP services | No[1] | – | – | – | – |
| | Yes | 0.5 (0.4, 0.7) | <0.001 | 0.7 (0.5, 0.9) | 0.001 |

RR: Risk ratio, CI: Confidence interval, ARR: Adjusted risk ratio, Growth monitoring and promotion, GMP; Non-governmental organization, NGO; and 1 = reference category.

[a]The Model included caregivers who heard about GMP or GMP cards, were members of an NGO, and lacked interest in GMP service.

## Discussion

This study assessed caregivers' knowledge, perceptions, utilization, and barriers related to growth monitoring and promotion (GMP) services in rural Bangladesh. It compared GMP services delivered at home with those delivered at health facilities. The study addresses a key evidence gap by providing one of the first comparative analyses of these two service delivery platforms. Findings showed that caregivers in home-based GMP areas had greater awareness of GMP, received more GMP cards, and were more likely to have their children measured and counseled. Overall GMP service utilization was 40% in home-based areas, compared to 0% in facility-based areas. This finding underscoring a stark contrast in service reach and engagement.

Our finding on low service utilization despite better knowledge of GMP in home-based settings is consistent with other LMIC settings [10,15,24–26]. Few studies have documented such findings, which is unique to this setting and contrast with the existing LMIC GMP program findings [9,26,27]. This contrasting finding may be due to the influence of differences in cultural beliefs, social norms, and contextual factors [24]. First, caregivers may perceive home-based GMP services as more accessible and convenient, particularly in rural settings where transportation barriers, long distances, or opportunity costs (e.g., loss of income or time away from household responsibilities) hinder facility visits [10,28,29]. Second, the longstanding presence of the NGO offering home-based services may have shaped caregiver preferences and trust, leading to reduced engagement with facility-based alternatives. Third, social norms and community practices—such as relying on community health workers rather than formal health facilities—may have further contributed to the non-use of facility-based GMP. Finally, weak accountability and inadequate supervision of GMP delivery at community clinics may result in poor service quality, discouraging caregivers from seeking these services [20]. Together, these contextual and structural factors likely contributed to the divergence between caregiver knowledge and service utilization, underscoring the need to strengthen facility-based GMP systems while understanding and addressing local behavioral drivers.

This study is the first to report on the absence of perceived benefit in facility-based sub-districts, which is unique as there is no government or research data available. Caregivers from both home- and facility-based GMP settings may prefer curative services over preventive services like GMP, a pattern commonly reported in LMIC settings [10,24,27,30]. It is possible that caregivers in our study context, similar to those in other studies, seek care only when their child is ill, primarily to receive treatment or access free medicines [10,28,31]. This behavior may reflect a broader trend in which preventive care is undervalued compared to curative interventions. Furthermore, existing literature suggests that caregivers are more likely to engage with health services that offer immediate and tangible benefits—such as vaccinations or financial incentives—which also apply to our study population [9–11,24,25,28,30]. These insights from the literature could help explain the low uptake of GMP services observed in our setting and underscore the importance of aligning service delivery with caregiver motivations and perceived benefits to enhance utilization.

Our findings indicated that caregivers' knowledge, perceptions, and barriers differed between home-based and facility-based GMP areas. GMP services were better utilized in home-based settings, likely due to the presence of NGO support, a trend also observed in other contexts [13,14,32]. However, we found that many caregivers had limited understanding of growth charts, consistent with previous studies that emphasized the need for clearer explanations by health workers [33,34]. In home-based areas, caregivers' awareness of GMP services appeared to be strongly linked to NGO efforts, while government outreach remained minimal [35]. Additionally, infrequent visits by government staff to health facilities may have contributed to poor utilization of facility-based GMP services, as also reported in similar settings. These patterns align with earlier research and suggest systemic gaps in engagement and service delivery, which may have limited caregivers' access to and understanding of GMP. We speculate that the insufficient supply of GMP cards in government facilities—reported by caregivers and observed during our field visits—may have been a key structural barrier, as echoed in the literature [10,24,33,36,37]. This shortage could stem from disrupted supply chains, inadequate funding, or low prioritization of GMP within the broader health system, ultimately limiting caregivers' ability to track and engage with their child's growth [36]. Other contextual barriers, such as facility closures and competing household responsibilities, may also have contributed to the low uptake of GMP services, a trend previously documented in LMICs [10,25].

To address these issues, systemic reforms should prioritize strengthening the government's supply chain to ensure the consistent availability of GMP cards and essential materials at health facilities. In parallel, enhanced training and accountability mechanisms for frontline health workers—along with targeted community outreach strategies, could help bridge information gaps and promote sustained engagement with GMP services. Finally, integrating caregiver education with routine health visits and introducing context-specific incentives may further improve utilization and impact. These findings collectively highlight the need for GMP programs that are better aligned with caregivers' preferences, behaviors, and the structural realities of their environments [8].

Home-based GMP services, despite their availability, exhibited low utilization rates, suggesting that accessibility alone may not drive engagement. Factors such as awareness, perceived benefits, and trust in service providers played a role in shaping caregivers' decisions. The observed differences in service utilization between home- and facility-based areas suggest the importance of tailoring program strategies to address setting-specific barriers and facilitators. For example, improved communication strategies are essential to enhance caregivers' understanding of growth charts—particularly explaining what the charts signify, how to interpret the colored zones, and what actions should be taken based on a child's growth trajectory [33,38]. Efforts should also ensure that all GMP components—such as weight monitoring, chart interpretation, and counseling—are consistently delivered during home visits. This could include structured home-visit checklists, routine supervision, and refresher training for frontline health workers. Barriers such as over-reliance on NGOs and limited government outreach reduce caregivers' awareness and access to GMP services. Addressing these challenges may require ensuring a consistent supply of GMP cards through improved supply chain coordination and introducing community-based demonstrations.

Strengthening partnerships between the government, NGOs, and community organizations is key to improving home-based GMP service utilization. Integrating NGO-led initiatives into the public health system could enhance the continuity and reach of services. This could involve joint training of health workers, shared protocols, and coordinated referral systems between home- and facility-based services. The success of NGO-supported, home-based GMP highlights the potential of community-based approaches, but scaling these models requires embedding them within national health systems. A hybrid approach, combining NGO innovations with government structures, could be effective, with shared workforce development and unified monitoring tools. To assess the feasibility of this hybrid government-NGO model, cost-effectiveness analyses should compare home-based and facility-based GMP delivery costs and outcomes. Evidence showed home-based GMP can become cost-effective when integrated with existing health platforms [13,38,39]. Though start-up costs may be high, integration with existing services and use of digital tools can improve cost-effectiveness and scalability over time [39].

The validity and reliability of the study findings may have been affected by several factors. Only one NGO provided home-based GMP services, and its longstanding presence in the community could have influenced caregivers' service uptake. However, similar sociodemographic characteristics among participants have likely reduced this bias. Despite the NGO's role and the accessibility of home-based services, utilization remained low, suggesting limited influence. We exclude children with chronic illness, severe malnutrition, and twins, which may introduce selection bias and limit generalizability. No data were collected on excluded children, so their impact on findings and GMP utilization patterns could not be assessed. Twins were excluded to minimize information bias. Though the study period was short, it still offers important insights into GMP utilization in rural settings. The study was conducted in a few rural subdistricts, excluding urban areas, which may limit the representativeness of national GMP patterns. However, two recent studies from South Asia and Africa reported no urban–rural differences in GMP utilization [25,28]. The cross-sectional design limits causal inference between caregivers' utilization and associated factors. The absence of longitudinal follow-up also prevents understanding of longer-term impacts. Facility-based GMP services were not utilized at all, preventing analysis of predictors in that group. Differences in baseline characteristics between groups may have contributed to outcome differences despite statistical adjustments. Thus, results should be interpreted with caution. Still, the study contributes novel evidence from a context where comparable government or research data are lacking [1].

Future research should explore these factors further to provide a more comprehensive understanding of GMP utilization dynamics. In particular, studies should explore sociocultural and behavioral influences on caregivers' decisions—such as preferences for curative over preventive care and trust in various service providers. Further investigation is also needed into the effectiveness of health worker communication strategies, the role of incentives, and the influence of service accessibility factors like distance, cost, and competing time demands. Comparative research across urban and rural contexts would also be valuable for identifying context-specific barriers and informing the design of more tailored, equitable GMP interventions. While this study did not assess long-term outcomes, prior evidence from LMICs suggests that consistent GMP engagement—particularly when combined with responsive counseling—can lead to earlier identification of growth faltering, improved caregiver feeding practices, and ultimately better child nutrition, development outcomes and productivity [11,28,39]. Longitudinal research in Bangladesh is needed to explore whether improved service utilization in home-based settings translates into sustained improvements in child growth and development metrics.

## Conclusion

This study provides important insights into the utilization dynamics of Growth Monitoring and Promotion (GMP) services in rural Bangladesh, highlighting distinct disparities between home-based and facility-based service delivery models. While home-based services showed relatively better utilization and caregiver knowledge, overall engagement remained suboptimal, pointing to deeper structural, behavioral, and systemic challenges. The complete absence of utilization in facility-based settings underscores the urgent need for strategic reforms that address supply chain issues, caregiver perceptions, service accessibility, and trust in service providers.

Our findings emphasize that improving GMP uptake requires more than availability—it necessitates the thoughtful integration of community-based strategies, caregiver education, consistent health worker engagement, and system-level coordination between NGOs and government services. While the study is limited by its cross-sectional design, rural-only focus, and short duration, it contributes novel evidence on GMP service delivery gaps and caregiver preferences in Bangladesh. Future research should build on these findings through longitudinal and intervention-based studies to identify sustainable approaches for improving GMP coverage and impact, particularly among underserved populations. Addressing sociocultural norms, strengthening facility-based systems, and aligning services with caregiver motivations will be critical to ensuring that GMP fulfills its preventive health potential.

## Supporting information

**S1 File. Details on statistical analysis.**
(DOCX)

**S2 File. Study questionnaire in English.**
(DOCX)

**S3 File. Study data set.**
(XLS)

**S4 File. Qualitative study tools.**
(DOCX)

## Acknowledgments

The study co-authors acknowledged the use of ChatGPT 3.5, developed by OpenAI, to edit the English language. The study team also acknowledges the role of two qualitative researchers, Ms. Ishrat Jahan and Dr. Ziaul Islam, for their contribution to qualitative data collection and analysis.

## Author contributions

**Conceptualization:** Muttaquina Hossain, Tahmeed Ahmed.

**Data curation:** Muttaquina Hossain.

**Formal analysis:** Muttaquina Hossain, Md. Ahshanul Haque.

**Funding acquisition:** Muttaquina Hossain, Tahmeed Ahmed.

**Investigation:** Muttaquina Hossain.

**Methodology:** Muttaquina Hossain.

**Project administration:** Muttaquina Hossain.

**Resources:** Muttaquina Hossain, Tahmeed Ahmed.

**Software:** Md. Ahshanul Haque.

**Supervision:** Muttaquina Hossain, Abu Syed Golam Faruque.

**Validation:** Muttaquina Hossain.

**Visualization:** Muttaquina Hossain.

**Writing – original draft:** Muttaquina Hossain, Per Ashorn.

**Writing – review & editing:** Muttaquina Hossain, Md. Ahshanul Haque, Abu Syed Golam Faruque, Tahmeed Ahmed, Ulla Ashorn, Per Ashorn.

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
