## [Decision Letter · Decision Letter 0]

3 Jan 2025

PONE-D-24-20319Caregiver utilization patterns in home-based and facility-based growth monitoring and promotion programs: A comparative analysisPLOS ONE

Dear Dr. Hossain,

Thank you for submitting your manuscript to PLOS ONE. After careful consideration, we feel that it has merit but does not fully meet PLOS ONE’s publication criteria as it currently stands. Therefore, we invite you to submit a revised version of the manuscript that addresses the points raised during the review process.

We look forward to receiving your revised manuscript.

Kind regards,

Zeheng Wang

Academic Editor

PLOS ONE

Journal Requirements:

4. We note you have included a table to which you do not refer in the text of your manuscript. Please ensure that you refer to Table 4 in your text; if accepted, production will need this reference to link the reader to the Table.

Reviewers' comments:

Reviewer's Responses to Questions

**Comments to the Author**

1. Is the manuscript technically sound, and do the data support the conclusions?

Reviewer #1: Yes

Reviewer #2: Yes

Reviewer #3: Yes

Reviewer #4: Yes

2. Has the statistical analysis been performed appropriately and rigorously? 

Reviewer #1: I Don't Know

Reviewer #2: Yes

Reviewer #3: Yes

Reviewer #4: Yes

3. Have the authors made all data underlying the findings in their manuscript fully available?

Reviewer #1: Yes

Reviewer #2: Yes

Reviewer #3: Yes

Reviewer #4: Yes

4. Is the manuscript presented in an intelligible fashion and written in standard English?

Reviewer #1: Yes

Reviewer #2: Yes

Reviewer #3: Yes

Reviewer #4: Yes

5. Review Comments to the Author

Reviewer #1: The study investigated the use of growth monitoring and promotion services in rural Bangladesh, comparing home-based and facility-based programs. The primary goal was to understand the factors influencing caregiver engagement with GMP services. The article's manuscript and statistical presentation of some of the major data appear to be relevant to a community-based study, particularly when considering the use of growth monitoring and promotion services. Its presentation and findings are satisfactory enough for publication. Therefore, I recommend acceptance of this manuscript, but with minor corrections.

The manuscript has used abbreviations inconsistently.

Line No. 14: Replace the GMP abbreviation with Growth Monitoring and Promotion (GMP).

Line No. 303: Growth Monitoring and Promotion Should be replaced with the GMP abbreviation.

Line Nos. 114, 259, and 309: the abbreviations GM for Growth Monitoring may not be required.

Reviewer #2: In addition to comments and feedbacks provided in reviewed manuscript attached, I want to mention that operational definitions viz wealth quintile, service utilization index etc should be added in method section. Some discussion is required like why facility based GMP utilization is almost zero, why home based service utilization is also lower than expectation ( around 40% only) etc

Reviewer #3: This is a very important and little studied topic.

Some points that I believe would be worth discussing further:

- The study is limited to rural sub-districts, not covering urban scenarios that may present different barriers and contexts.

- The absence of longitudinal data prevents the assessment of long-term impact.

- The gap in the provision of GMP cards in government facilities highlights a structural problem that could be discussed more strongly.

- Specific strategies to better integrate NGO initiatives into the public health system could be better described.

- Qualitative analysis presents data that could be further explored.

Reviewer #4: This study evaluates and compares utilization patterns and barriers to caregiver engagement in Growth Monitoring and Promotion (GMP) services delivered via home-based (3 sub-districts) and facility-based (3 sub-districts) mechanisms in Mymensingh district, Bangladesh. The research aims to identify efficient and effective public health delivery models in a resource-limited setting of a low- and middle-income country. While well-written, the proposal could benefit from addressing the following comments:

• Title: The primary focus of the study is to assess and compare GMP service utilization and identify the barriers; however, the title focuses on the caregiver utilization. Consider the following alternative titles:

o "Disparities in Utilization and Barriers to Caregiver Engagement in Growth Monitoring and Promotion Services: A Comparative Study of Home-Based and Facility-Based Programs."

o "Disparities and Barriers in Growth Monitoring and Promotion Services: Home-Based vs. Facility-Based Programs."

o "Growth Monitoring and Promotion Program Services Utilization Patterns Between Home-Based and Facility-Based Delivery Methods: A Comparative Analysis."

o "Comparative Utilization Patterns of Growth Monitoring and Promotion Services: Home-Based Versus Facility-Based Delivery Methods."

• Page 2 Line 14: Expand the abbreviation GMP at its first use in the abstract.

• Page 3 Line 38: “Conclusion: GMP service utilization remains low in some parts of rural Bangladesh”.

o Compare and contrast the sociodemographic characteristics of the Mymensingh district with other similar districts and national metrics in Bangladesh.

o Provide a clear rationale for selecting Mymensingh for this comparative study, apart from the presence of NGO services.

o Analyze and compare the characteristics of sub-districts offering home-based versus facility-based GMP services.

• Page 6 Line 104-108:

o Justify the exclusion of children with acute and chronic illnesses, stunting, wasting, underweight, or those not enrolled in similar programs, as these groups might lack access to or awareness of GMP services.

o Indicate the proportion of children excluded and assess the impact on study findings and GMP service utilization metrics.

o Discuss potential selection biases introduced by these exclusions, the rate of GMP service utilization among excluded children.

o Explain the bias introduced by including twins in the study and its implications for the results.

• Page 7 Line 111-112: “The components of GMP are intended, not documented, actions.” Rewrite the sentence with clarity.

• Page 7 Line 115: Could the poor health outcomes of excluded children be attributed to a lack of access to GMP services?

• Page 9 Line 163: What proportion of responses were identified as incorrect?

• Page 10 Line 190-192: Were questionaries in Bangla validated? If yes, please state it in the paper.

• Page 10 Line 193: The abbreviation CC (Community Clinic) was already introduced earlier in line 119.

• Page 11 Line 207-208: “Only some relevant participants were interviewed for the qualitative part.” Define "relevant participants" in the qualitative analysis and explain how they differ from those in the quantitative survey. Show sociodemographic characteristics of both groups.

• Page 11 Line 208-209: “The number of qualitative interviews was decided following an iterative process of achieving information saturation.” Explain the ‘iterative process’ and ‘information saturation’ for non-technical readers.

• Page 12 Line 224-225: Expand the abbreviation LAZ, WAZ, and WHZ at their first use.

• Page 14 Line 235: It appears unrealistic that only 0.4% caregiver would have heard of GMP cards. Please provide government data and other research that have reported GMP card awareness in the facility-based GMP services in Bangladesh.

• Page 14 Line 240-241:

o The reported awareness (0.4%) of GMP cards seems unrealistically low. Provide government and other research data for comparison, particularly for facility-based GMP services in Bangladesh.

o Additionally, clarify the definition of “beneficial” and address the absence of perceived benefit in facility-based sub-districts using government and other research data.

• Table 1: Were sociodemographic characteristics statistically different? Include P-value.

• Figure 1: “Selection and matching…”. Was matching performed? If matching was performed between home-based and facility-based cohorts, provide methodological details.

• Add the following supplementary materials:

o Include details of statistical analyses, such as normality tests and full models with covariates.

o Add the questionnaire (with English translation) to aid reader comprehension of questions and outcome metrics.

6. PLOS authors have the option to publish the peer review history of their article (what does this mean? ). If published, this will include your full peer review and any attached files.

**Do you want your identity to be public for this peer review?** For information about this choice, including consent withdrawal, please see our Privacy Policy .

Reviewer #1: No

Reviewer #2: **Yes: ** Krishna Deo Yadav

Reviewer #3: No

Reviewer #4: **Yes: ** Sunil Swami

---

## [Author Response · Author response to Decision Letter 1]

9 Feb 2025

6 February 2025

Zeheng Wang

Academic Editor

PLOS ONE

California, USA

RE: PONE-D-24-20319: Growth monitoring and promotion program services utilization patterns between home-based and facility-based delivery methods: a comparative analysis

We thank you for your response to our manuscript. Please find attached a revised version, in response to your invitation to resubmit the paper to PLOS ONE if we felt that we could satisfactorily address the concerns of the reviewers. We have substantially modified the manuscript, taking into account the comments from the editorial office and the four reviewers.

Our response to the comments is presented using the following structure: The original editorial / referee comment is copied into this letter in italics. Our response to the comment is written below in normal text format. In case a revision to the manuscript text has been made, we give the appropriate page number(s). Changes in the text have been indicated using yellow text highlighting in the manuscript and copied into this letter in the same format. The highlighted text file has been uploaded into your electronic system as a “Revised Manuscript with Track Changes”.

Responses to the comments made by the academic editor:

Our response: Thank you for your suggestion. We reformatted the manuscript by following the PLOS ONE’s style requirements including those for file naming.

Our response: We removed funding information from the manuscript as suggested.

Our response: We would like to share the data as Supporting information file reads as (line 596):

S3 File. Study data set

4. We note you have included a table to which you do not refer in the text of your manuscript. Please ensure that you refer to Table 4 in your text; if accepted, production will need this reference to link the reader to the Table.

Our response: We now include Table 4 inside the text (lines 306, 314).

“Four major barriers were identified as reasons for caregivers' underutilization of GMP services. Among these, caregivers' lack of knowledge about the availability of GMP services at the nearest health facility was a significant barrier in both areas (Table 4).”

More caregivers from home-based GMP service areas reported closure of health facilities/absence of service providers (1%) and being preoccupied with household work (4%) as significant barriers to accessing and utilizing GMP services, compared to facility-based GMP service areas (0.2% and 0.3%, respectively) (Table 4).”

Responses to the comments made by reviewer # 1:

Reviewer #1: The study investigated the use of growth monitoring and promotion services in rural Bangladesh, comparing home-based and facility-based programs. The primary goal was to understand the factors influencing caregiver engagement with GMP services. The article's manuscript and statistical presentation of some of the major data appear to be relevant to a community-based study, particularly when considering the use of growth monitoring and promotion services. Its presentation and findings are satisfactory enough for publication. Therefore, I recommend acceptance of this manuscript, but with minor corrections.

1. The manuscript has used abbreviations inconsistently.

Line No. 14: Replace the GMP abbreviation with Growth Monitoring and Promotion (GMP).

Our response: Thank you for pointing this out. We provided the full form of GMP (lines 14-15).

“In Bangladesh, utilization of government health facility-based growth monitoring and promotion (GMP) services is notably low, prompting non-governmental organizations (NGOs) to collaborate with the government to enhance GMP service utilization through home-based delivery.”

2. Line No. 303: Growth Monitoring and Promotion Should be replaced with the GMP abbreviation.

Our response: We replaced the full form with GMP abbreviation, as suggested (line 349).

“The study aimed to assess caregivers' knowledge, perception, utilization, and barriers to accessing GMP services in rural Bangladesh, comparing home-based and facility-based settings.”

3. Line Nos. 114, 259, and 309: the abbreviations GM for Growth Monitoring may not be required.

Our response: We kept the full form “growth monitoring” and removed GM abbreviation, as suggested (lines 137, 301, 302).

Responses to the comments made by reviewer # 2

Reviewer #2: In addition to comments and feedbacks provided in reviewed manuscript attached, I want to mention that operational definitions viz wealth quintile, service utilization index etc should be added in method section. Some discussion is required like why facility based GMP utilization is almost zero, why home based service utilization is also lower than expectation ( around 40% only) etc

Our response: Thank you so much for your valuable comments and guidance. We addressed all your comments inside the manuscript, comment boxes as suggested and highlighted them.

• The definition of wealth index revised text in the methods section reads as follows (lines 188-190):

“A wealth index was constructed from household assets using a composite score to classify socioeconomic status (19). Households were grouped into three categories: rich, middle, and poor, reflecting relative differences in asset ownership.”

• We added the term “operational definition” for service utilization index in methods section reads as follows (line 157):

“Operational definition: Caregivers' GMP service utilization was defined as whether they received GMP services at home or went to the facility for the service even once during the child’s lifetime during the survey”

• We provided the reason in discussion section on “why facility based GMP utilization is almost zero, why home based service utilization is also lower than expectation” reads as follows (lines 359-368):

“The stark contrast in GMP service utilization, with home-based services at 40% and facility-based services at 0%, underscores critical barriers in accessing and delivering care. While home-based services perform better, their utilization remains lower than expected due to irregular or insufficient visits by health workers, limited caregiver awareness of GMP’s importance, and competing priorities or time constraints (21). In contrast, the zero utilization of facility-based GMP highlights additional challenges such as limited accessibility to health facilities, lack of awareness or trust in facility-based services, and caregivers’ preference for more convenient or culturally appropriate home-based interventions (22). Moreover, the absence of facility-based utilization may reflect systemic gaps in infrastructure, staffing, and health system outreach efforts (6).

Responses to the comments made by reviewer # 3:

Reviewer #3: This is a very important and little studied topic.

Some points that I believe would be worth discussing further:

1. The study is limited to rural sub-districts, not covering urban scenarios that may present different barriers and contexts.

Our response: We agree with your concern. We already mentioned this as limitation in the discussion section (lines 441-443):

“Firstly, it was conducted in only a few subdistricts of rural Bangladesh, excluding urban areas, potentially compromising the representation of the country's overall GMP utilization scenario.”

2. The absence of longitudinal data prevents the assessment of long-term impact.

Our response: Yes, we agree with your comment and added the following sentence in discussion section, read as follows (lines 444-445):

“Thirdly, the absence of longitudinal data prevents the assessment of long-term impact.”

3. The gap in the provision of GMP cards in government facilities highlights a structural problem that could be discussed more strongly.

Our response: We edited as suggested in the discussion section, read as follows (lines 407-414):

“The insufficient supply of GMP cards in government facilities significantly contributes to their non-utilization and highlights a critical structural issue within the health system [26, 31, 32]. This shortfall stems from disrupted supply chains, limited funding, or inadequate prioritization of GMP services, ultimately hindering caregivers’ ability to track and engage with child growth monitoring [32]. Barriers like health facility closures and household responsibilities also explain low utilization rates [6]. To address this, systemic reforms are needed to ensure a reliable supply of GMP cards, as they are essential tools for promoting effective service delivery and improving child health outcomes.”

4. Specific strategies to better integrate NGO initiatives into the public health system could be better described.

Our response: We now added strategies that could help to integrate NGO initiatives into the public health system in discussion section, read as follows (lines 422-427):

“To improve GMP utilization, specific strategies for integrating NGO initiatives into the public health system should focus on strengthening coordination and resource sharing. This includes joint training of staff, aligning service delivery goals, and establishing referral and follow-up mechanisms between home-based and facility-based services [33, 34]. Such integration can bridge service gaps, enhance caregiver awareness, and improve access to GMP services.”

5. Qualitative analysis presents data that could be further explored.

Response: Our response: We have added information from qualitative analysis in the results section (lines 320-324):

“Caregivers from facility-based GMP areas never received GMP cards because of lack of supply of GMP cards in CCs from the government. Interviews with government stakeholders revealed the lack of allocation of government funds to print and supply GMP cards to the CCs during the study period. The NGO area caregivers got GMP cards as the NGO printed out and distributed government approved GMP cards using their own funds.”

Responses to the comments made by reviewer # 4:

Reviewer #4: This study evaluates and compares utilization patterns and barriers to caregiver engagement in Growth Monitoring and Promotion (GMP) services delivered via home-based (3 sub-districts) and facility-based (3 sub-districts) mechanisms in Mymensingh district, Bangladesh. The research aims to identify efficient and effective public health delivery models in a resource-limited setting of a low- and middle-income country. While well-written, the proposal could benefit from addressing the following comments:

• Title: The primary focus of the study is to assess and compare GMP service utilization and identify the barriers; however, the title focuses on the caregiver utilization. Consider the following alternative titles:

o "Disparities in Utilization and Barriers to Caregiver Engagement in Growth Monitoring and Promotion Services: A Comparative Study of Home-Based and Facility-Based Programs."

o "Disparities and Barriers in Growth Monitoring and Promotion Services: Home-Based vs. Facility-Based Programs."

o "Growth Monitoring and Promotion Program Services Utilization Patterns Between Home-Based and Facility-Based Delivery Methods: A Comparative Analysis."

o "Comparative Utilization Patterns of Growth Monitoring and Promotion Services: Home-Based Versus Facility-Based Delivery Methods."

Our response: Thank you for suggesting 4 options for a title. We picked the 3rd one and revised our title as “Growth Monitoring and Promotion Program Services Utilization Patterns Between Home-Based and Facility-Based Delivery Methods: A Comparative Analysis."

• Page 2 Line 14: Expand the abbreviation GMP at its first use in the abstract.

Our response: We edited as suggested, read as follows (line 14-15):

“Background: In Bangladesh, utilization of government health facility-based growth monitoring and promotion (GMP) services is notably low, prompting non-governmental organizations (NGOs) to collaborate with the government to enhance GMP service utilization through home-based delivery.”

• Page 3 Line 38: “Conclusion: GMP service utilization remains low in some parts of rural Bangladesh”.

Our response: We revised the sentence and now read as following (line 38):

“Conclusion: GMP service utilization remains low in rural Mymensingh district of Bangladesh.”

o Compare and contrast the sociodemographic characteristics of the Mymensingh district with other similar districts and national metrics in Bangladesh.

Our response: We added the information, read as following (lines 94-95):

“Mymensingh is similar to other Bangladeshi districts with its predominantly rural, agrarian population, along with comparable literacy rates and demographic patterns.”

o Provide a clear rationale for selecting Mymensingh for this comparative study, apart from the presence of NGO services.

Our response: This following information has been added in the methods section (lines 95-98):

“We selected the Mymensingh district based on the following criteria: (A) time of implementation of the programme matching with the study timeline; (B) staff and logistics in CCs supported by NGO; and (C) an adequate number of CCs in subdistricts to satisfy the sample size [13].”

o Analyze and compare the characteristics of sub-districts offering home-based versus facility-based GMP services.

Our response: We added the information in the results section reads as follows (lines 268-269) and uploaded the table as supplementary file:

“The socio-demographic characteristics of the six sub-districts in 2 GMP areas were mostly comparable (S1 file).”

• Page 6 Line 104-108:

o Justify the exclusion of children with acute and chronic illnesses, stunting, wasting, underweight, or those not enrolled in similar programs, as these groups might lack access to or awareness of GMP services.

Our response: We revised as suggested, read as follows (lines 125-128):

“Eligibility criteria encompassed children free from known acute/chronic illnesses, severe stunting, wasting, or underweight due to ethical obligation as the study would not provide treatment or management, not enrolled in similar programs, and whose caregivers provided consent.”

o Indicate the proportion of

---

## [Decision Letter · Decision Letter 1]

11 Mar 2025

PONE-D-24-20319R1Growth monitoring and promotion program services utilization patterns between home-based and facility-based delivery methods: a comparative analysisPLOS ONE

Dear Dr. Hossain,

Thank you for submitting your manuscript to PLOS ONE. After careful consideration, we feel that it has merit but does not fully meet PLOS ONE’s publication criteria as it currently stands. Therefore, we invite you to submit a revised version of the manuscript that addresses the points raised during the review process.

We look forward to receiving your revised manuscript.

Kind regards,

Zeheng Wang

Academic Editor

PLOS ONE

Additional Editor Comments:

Old reviewers not available. New reviewers pointed out that the manuscript merits publication upon another Major Revision.

Reviewers' comments:

Reviewer's Responses to Questions

**Comments to the Author**

1. If the authors have adequately addressed your comments raised in a previous round of review and you feel that this manuscript is now acceptable for publication, you may indicate that here to bypass the “Comments to the Author” section, enter your conflict of interest statement in the “Confidential to Editor” section, and submit your "Accept" recommendation.

Reviewer #4: All comments have been addressed

Reviewer #5: All comments have been addressed

Reviewer #6: All comments have been addressed

Reviewer #7: All comments have been addressed

Reviewer #8: (No Response)

2. Is the manuscript technically sound, and do the data support the conclusions?

Reviewer #4: Yes

Reviewer #5: Yes

Reviewer #6: Partly

Reviewer #7: Partly

Reviewer #8: Partly

3. Has the statistical analysis been performed appropriately and rigorously? 

Reviewer #4: Yes

Reviewer #5: Yes

Reviewer #6: I Don't Know

Reviewer #7: Yes

Reviewer #8: Yes

4. Have the authors made all data underlying the findings in their manuscript fully available?

Reviewer #4: Yes

Reviewer #5: (No Response)

Reviewer #6: Yes

Reviewer #7: Yes

Reviewer #8: (No Response)

5. Is the manuscript presented in an intelligible fashion and written in standard English?

Reviewer #4: Yes

Reviewer #5: Yes

Reviewer #6: Yes

Reviewer #7: Yes

Reviewer #8: (No Response)

6. Review Comments to the Author

Reviewer #4: (No Response)

Reviewer #5: Strengths of the Manuscript

The study addresses an important public health issue related to child growth monitoring and promotion (GMP) services in Bangladesh. The comparison between home-based and facility-based service utilization provides valuable insights for health policy and program implementation. The study is data-driven, with a substantial sample size (N=3038) and a mixed-method approach, which enhances the credibility of the findings. The comparative cross-sectional design is well-suited for assessing utilization patterns. The quantitative surveys and qualitative interviews ensure a comprehensive assessment of caregiver engagement. The use of multiple logistic regression models and risk ratio (RR) calculations is appropriate. The manuscript effectively describes statistical adjustments. The findings have policy implications, which could improve GMP service delivery in LMICs.

Areas for Improvement

The manuscript is well-written but contains redundancies in some sections. The introduction could be more concise, avoiding repetitive statements about the importance of GMP. Some statistical results could be integrated into the discussion to provide stronger interpretation of findings and improve the depth of the discussion.

While the discussion section provides good insights, it could be expanded to:

o Compare findings with other countries that have implemented home-based GMP.

o Elaborate on potential policy changes to integrate NGO-driven GMP services into government programs.

o Address the scalability and sustainability of home-based GMP programs.

o Discuss the long-term health outcomes of children engaged in these services.

4. Addressing Potential Biases

• The study excludes urban areas, which limits generalizability. Although this is mentioned as a limitation, a comparison with urban settings from prior research could strengthen the discussion.

• Selection bias due to the exclusion of children with severe malnutrition could affect findings. Further discussion on this would improve credibility.

5. Visual Presentation of Data

• Figures and Tables:

o The flow diagram (Figure 1) could be more visually engaging.

o Tables 3-5 should be condensed where possible to improve readability.

o A graphical summary of key results (e.g., a bar chart showing GMP utilization disparities) would enhance clarity.

6. Policy & Implementation Implications

• The study suggests NGO-driven GMP services improve utilization, but how can this be implemented at scale?

• Policy recommendations should be expanded:

o Should the government adopt a hybrid approach with NGO support?

o What cost-effectiveness considerations exist for scaling up home-based GMP?

Reviewer #6: The paper presents an interesting scenario in rural Bangladesh where there is poor use of Goverment Servcies for the monitoring of children post-natally and the authors attempted to understand why going into the research thinking that the main reason might be the fact that there was a lack of knowledge that these services were offered. This in itself makes for an interesting study.

While the methods presented are very detailed, I have some concerns about whether the authors of this paper are trying to report too much in a single study for example I am not sure as to why there was a need to report mean birth weight and birth length in this study as it did not directly lend to the objectives of 1st determining whether there was a gap and 2. establishing the barriers. I suggest only reporting data that directly answers these two objectives. Further than knowledge, I am not clear what other barriers were explored and how they were explored. There was mention of thematic analysis but there is no reporting of the data and outcomes of the thematic analysis using accepted guidleines.

Reviewer #7: 1. The authors have sufficiently addressed the feedback from the previous review round, and I believe this manuscript is now suitable for publication.

2. The manuscript presents a technically solid piece of scientific research; however, the conclusions chapter requires expansion.

3.The statistical analysis has been conducted appropriately and rigorously.

4.The authors have made all the data supporting their manuscript's findings fully accessible.

5.The manuscript is presented clearly and is written in standard English.

Reviewer #8: (No Response)

7. PLOS authors have the option to publish the peer review history of their article (what does this mean? ). If published, this will include your full peer review and any attached files.

**Do you want your identity to be public for this peer review?** For information about this choice, including consent withdrawal, please see our Privacy Policy .

Reviewer #4: **Yes: ** Sunil Swami

Reviewer #5: **Yes: ** Negar Yousefzadeh, NIHR Innovation Observatory, Population Health Sciences Institute, Newcastle University, Newcastle Upon Tyne, UK

Reviewer #6: No

Reviewer #7: No

Reviewer #8: **Yes: ** Shabnam Varmazyari, DDSS-MPH, Research Assistant at Tehran University of Medical Sciences

---

## [Author Response · Author response to Decision Letter 2]

17 Apr 2025

Responses to the comments made by reviewer # 5:

The manuscript is well-written but contains redundancies in some sections. The introduction could be more concise, avoiding repetitive statements about the importance of GMP. Some statistical results could be integrated into the discussion to provide stronger interpretation of findings and improve the depth of the discussion.

Our response: We appreciate your kind observation. We now have revised the introduction, reads as follows (lines 60-71):

“Growth Monitoring and Promotion (GMP) is a key part of child health programs that involve regularly measuring a child’s growth to detect early signs of growth faltering and support timely interventions. The GMP program is designed to include regular check-ups for child growth and happens every three months at health facilities. In an ideal GMP program, Health workers are advised to counsel mothers and caregivers on their children's nutrition, measure the weight, length, or mid-upper arm circumference (MUAC) of all under-2-year-old children, and plot the results on a growth monitoring chart in the GMP card. If a health worker identifies a child as acutely ill or not growing well during GMP, he/she should refer the child to the nearest medical/treatment center [1]. This approach aims to foster communication and interaction with caregivers, promoting optimal child development [2]. UNICEF’s introduction of the promotion component in the mid-1980s strengthened GMP, making it a key part of nutrition programs for managing child malnutrition in low- and middle-income countries [3].”

The discussion section already contains results (lines 485-492):

“Sixty-one percent of caregivers received GMP cards at home-based GMP, whereas none did in facility-based regions due to a lack of supply. This explains why only 0.4% of caregivers have heard of GMP cards in facility-based GMP areas. ………More children were measured for weight and length at home-based services, and their caregivers received growth monitoring-specific counseling compared to facilities (81% vs. 21%; 3.4% vs. 0.6%; and 55% vs. 05%, respectively). GMP service utilization was higher among caregivers at home compared to facility-based GMP services (40% vs. 0%, respectively).”

While the discussion section provides good insights, it could be expanded to:

o Compare findings with other countries that have implemented home-based GMP.

Our response: Edited as suggested, reads as follows (lines 521-537):

“Our finding on low service utilization despite better knowledge of GMP in home-based settings is consistent with other LMIC settings [10, 16, 25, 26, 29]. Few studies have documented such findings, which is unique to this setting and contrast with the existing LMIC GMP program findings [9, 28, 29].This contrasting finding may be due to the influence of differences in cultural beliefs, social norms, and contextual factors [26]. First, caregivers may perceive home-based GMP services as more accessible and convenient, particularly in rural settings where transportation barriers, long distances, or opportunity costs (e.g., loss of income or time away from household responsibilities) hinder facility visits [10, 30]. Second, the longstanding presence of the NGO offering home-based services may have shaped caregiver preferences and trust, leading to reduced engagement with facility-based alternatives. Third, social norms and community practices—such as relying on community health workers rather than formal health facilities—may have further contributed to the non-use of facility-based GMP. Finally, weak accountability and inadequate supervision of GMP delivery at community clinics may result in poor service quality, discouraging caregivers from seeking these services [20].Together, these contextual and structural factors likely contributed to the divergence between caregiver knowledge and service utilization, underscoring the need to strengthen facility-based GMP systems while understanding and addressing local behavioral drivers.”

o Elaborate on potential policy changes to integrate NGO-driven GMP services into government programs

Our response: Edited as suggested, reads as follows [lines 593-606]:

“Strengthening partnerships between the government, NGOs, and community organizations is key to improving home-based GMP service utilization. Integrating NGO-led initiatives into the public health system could enhance the continuity and reach of services. This could involve joint training of health workers, shared protocols, and coordinated referral systems between home- and facility-based services. The success of NGO-supported, home-based GMP highlights the potential of community-based approaches, but scaling these models requires embedding them within national health systems. A hybrid approach, combining NGO innovations with government structures, could be effective, with shared workforce development and unified monitoring tools. To assess the feasibility of this hybrid government-NGO model, cost-effectiveness analyses should compare home-based and facility-based GMP delivery costs and outcomes. Evidence showed home-based GMP can become cost-effective when integrated with existing health platforms [14, 39, 40]. Though start-up costs may be high, integration with existing services and use of digital tools can improve cost-effectiveness and scalability over time [40].”

o Address the scalability and sustainability of home-based GMP programs.

Our response: Edited as suggested, reads as follows [lines 578-592]:

“Home-based GMP services, despite their availability, exhibited low utilization rates, suggesting that accessibility alone may not drive engagement. Factors such as awareness, perceived benefits, and trust in service providers played a role in shaping caregivers' decisions. The observed differences in service utilization between home- and facility-based areas suggest the importance of tailoring program strategies to address setting-specific barriers and facilitators. For example, improved communication strategies are essential to enhance caregivers’ understanding of growth charts—particularly explaining what the charts signify, how to interpret the colored zones, and what actions should be taken based on a child’s growth trajectory [34, 39]. Efforts should also ensure that all GMP components—such as weight monitoring, chart interpretation, and counseling—are consistently delivered during home visits. This could include structured home-visit checklists, routine supervision, and refresher training for frontline health workers. Barriers such as over-reliance on NGOs and limited government outreach reduce caregivers' awareness and access to GMP services. Addressing these challenges may require ensuring a consistent supply of GMP cards through improved supply chain coordination and introducing community-based demonstrations.”

o Discuss the long-term health outcomes of children engaged in these services

Our response: Edited as suggested, reads as follows [lines 618-621]:

“While this study did not assess long-term outcomes, prior evidence from LMICs suggests that consistent GMP engagement—particularly when combined with responsive counseling—can lead to earlier identification of growth faltering, improved caregiver feeding practices, and ultimately better child nutrition, development outcomes and productivity [11, 13, 40]. Longitudinal research in Bangladesh is needed to explore whether improved service utilization in home-based settings translates into sustained improvements in child growth and development metrics.”

4. Addressing Potential Biases

• The study excludes urban areas, which limits generalizability. Although this is mentioned as a limitation, a comparison with urban settings from prior research could strengthen the discussion

Our response: Edited as suggested, read as follows [lines 623-627]:

“Firstly, it was conducted in only a few subdistricts of rural Bangladesh, excluding urban areas, potentially compromising the representation of the country's overall GMP utilization scenario, similar to other prior research studies from LMICs [13]. However, two recent studies conducted in South Asia and Africa showed no urban-rural differences in GMP utilization status [25, 30].”

• Selection bias due to the exclusion of children with severe malnutrition could affect findings Further discussion on this would improve credibility

Our response: Edited as suggested, reads as follows [lines 510-514]:

“Our study included only apparently healthy, disease-free children, which might introduce a selection bias and limit the generalizability of the findings to children with underlying health conditions, including severe malnutrition. We did not collect information from excluded children and thus could not assess the impact on study findings and the GMP service utilization matrix.”

5. Visual Presentation of Data

• Figures and Tables:

o The flow diagram (Figure 1) could be more visually engaging.

Our response: Edited as suggested, reads as follows:

o Tables 3-5 should be condensed where possible to improve readability.

Our response: Edited as suggested and aligned with reviewer SV’s edits, reads as follows:

Table 3. Levels of GMP service utilization among individuals in facility (N = 1,519) and home-based (N = 1,519) GMP programs (N = 1,519)

Indicators Home-based GMP N (%) Facility-based GMP N (%) RR (95% CI) p-value

Caregivers received a GMP card 926 (60.9) 0 (0) - -

The child was weighed at least once 1231 (81.0) 311 (20.5) 4.0 (3.6, 4.4) <0.001

The child length was measured at least once 51 (3.36) 09 (0.59) 5.7 (2.8, 11.5) <0.001

Caregivers got child growth monitoring-specific counseling 837 (55.2) 88 (5.4) 9.5 (7.7, 11.7) <0.001

The child received all major components of GMP (length, weight, and growth monitoring-specific counseling) altogether 30 (1.97) 0 (0) - -

Caregivers utilized GMP services 603 (39.7) 0 (0) - -

Growth monitoring and promotion, GMP; confidence interval, CI; Risk Ratio, RR.

Table 4: Barriers to GMP service utilization among individuals in the facility (N = 1,519) and home-based (N = 1,519) GMPa programs

Indicators Home-based GMP N (%)

Facility-based GMP N (%)

RR (95% CI) p-value

The caregiver was preoccupied with household chores and did not go for the GMP service 56 (3.7) 5 (0.3) 11.2 (4.5, 27.9) <0.001

The health facility was closed/no healthcare provider at the facility 14 (0.9) 3 (0.2) 4.6 (1.3, 16.2) 0.015

Caregivers' lack of interest in GMP services at an individual level 668 (44.0) 834 (54.9) 0.7 (0.6, 0.9) 0.006

Lack of knowledge of GMP service availability at the nearest health facility 798 (52.5) 919 (60.5) 0.9 (0.8, 0.9) <0.001

Growth monitoring and promotion, GMP; confidence interval, CI; Risk Ratio, RR.

aMultiple responses by caregivers.

Table 5. Predictors of GMP service utilization among caregivers at the home-based GMP programs

Characteristica Characteristic category RR (95% CI) p-value ARR (95% CI) p-value

Caregiver having heard about GMP or GMP card No1 - - - -

Yes 42.4 (20.3, 88.6) <0.001 37.4 (17.8, 78.5) <0.001

Caregiver being a member of an association/NGO/health program No1 - - - -

Yes 2.6 (2.1, 3.2) <0.001 1.3 (1.1, 1.5) 0.001

Caregiver's low interest in GMP services No1 - - - -

Yes 0.5 (0.4, 0.7) <0.001 0.7 (0.5, 0.9) 0.001

RR: Risk ratio, CI: Confidence interval, ARR: Adjusted risk ratio, Growth monitoring and promotion, GMP; Non-governmental organization, NGO; and 1 = reference category

aThe Model included caregivers who heard about GMP or GMP cards, were members of an NGO, and lacked interest in GMP service.

o A graphical summary of key results (e.g., a bar chart showing GMP utilization disparities) would enhance clarity.

Our response: Added as suggested (line 401), and uploaded as Fig 2:

Fig 2. Caregivers’ knowledge and level of GMP service utilization

6. Policy & Implementation Implications

• The study suggests NGO-driven GMP services improve utilization, but how can this be implemented at scale?

Our response: Edited as suggested (lines 594-601):

“Integrating NGO-led initiatives into the public health system could enhance the continuity and reach of services. This could involve joint training of health workers, shared protocols, and coordinated referral systems between home- and facility-based services. The success of NGO-supported, home-based GMP highlights the potential of community-based approaches, but scaling these models requires embedding them within national health systems. A hybrid approach, combining NGO innovations with government structures, could be effective, with shared workforce development and unified monitoring tools. ”

• Policy recommendations should be expanded:

o Should the government adopt a hybrid approach with NGO support?

Our response: Edited as suggested (lines 599-601):

“A hybrid approach, combining NGO innovations with government structures, could be effective, with shared workforce development and unified monitoring tools.”

o What cost-effectiveness considerations exist for scaling up home-based GMP?

Our response: Edited as suggested (lines 601-606):

“To assess the feasibility of this hybrid government-NGO model, cost-effectiveness analyses should compare home-based and facility-based GMP delivery costs and outcomes. Evidence showed home-based GMP can become cost-effective when integrated with existing health platforms [14, 39, 40]. Though start-up costs may be high, integration with existing services and use of digital tools can improve cost-effectiveness and scalability over time [40].”

Responses to the comments made by Reviewer #6:

While the methods presented are very detailed, I have some concerns about whether the authors of this paper are trying to report too much in a single study for example I am not sure as to why there was a need to report mean birth weight and birth length in this study as it did not directly lend to the objectives of 1st determining whether there was a gap and 2. establishing the barriers. I suggest only reporting data that directly answers these two objectives.

Our response: Thank you for your query and suggestion. We did not measure and report birth weight and birth length of the child. We reported child weight and length at enrolment to understand and compare any difference in anthropometric incidence of the children from home and facility-based GMP area (lines 357-360), the comparison was done as per suggestion from 1st round reviewer’s comment:

“Children's length-for-age Z-score (LAZ) and weight-for-age Z score (WAZ) was comparable among the groups. Children from home-based GMP area had significantly lower weight-for-length/height Z-score (WHZ) compared to facility-based area children (mean WHZ: -0.48 versus -0.4, p=0.04).”

Further than knowledge, I am not clear what other barriers were explored and how they were explored.

Our response: Our apologies for this confusion. Edited as suggested (lines 199-202):

“The caregivers’ barriers to access and utilization of GMP services were explored using a semi-structured survey questionnaire and during qualitative interviews. The survey questionnaire included multiple response options.”

There was mention of thematic analysis but there is no reporting of the data and outcomes of the thematic analysis using accepted guidelines.

Our response: Edited as suggested (lines 287-297, 322-340):

“The theoretical framework guiding this study is the Health Belief Model (HBM), which emphasizes individuals’ perceptions of health risks, the benefits of preventive actions, and the barriers to adopting these actions. In the context of this study, the HBM helped to understand caregivers' motivations and decision-making regarding GMP services. It informed the development of research questions related to perceived severity, susceptibility, benefits, and barriers to GMP service utilization. This framework also guided the identification of key themes during the qualitative analysis, focusing on the barriers caregivers face and the factors that influence their engagement with GMP services. By using the HBM, the study aimed to provide a structured understanding of caregivers' behaviors and attitudes toward health service utilization in rural Bangladesh.”

“Qualitative analysis involved thematic descriptions, analysis,

---

## [Decision Letter · Decision Letter 2]

29 Apr 2025

PONE-D-24-20319R2Growth monitoring and promotion program services utilization patterns between home-based and facility-based delivery methods: a comparative analysisPLOS ONE

Dear Dr. Hossain,

Thank you for submitting your manuscript to PLOS ONE. After careful consideration, we feel that it has merit but does not fully meet PLOS ONE’s publication criteria as it currently stands. Therefore, we invite you to submit a revised version of the manuscript that addresses the points raised during the review process.

We look forward to receiving your revised manuscript.

Kind regards,

Zeheng Wang

Academic Editor

PLOS ONE

Journal Requirements:

Additional Editor Comments:

The reviewers suggest that the manuscript can be published after a minor revision.

Reviewers' comments:

Reviewer's Responses to Questions

**Comments to the Author**

1. If the authors have adequately addressed your comments raised in a previous round of review and you feel that this manuscript is now acceptable for publication, you may indicate that here to bypass the “Comments to the Author” section, enter your conflict of interest statement in the “Confidential to Editor” section, and submit your "Accept" recommendation.

Reviewer #7: All comments have been addressed

Reviewer #8: All comments have been addressed

2. Is the manuscript technically sound, and do the data support the conclusions?

Reviewer #7: Yes

Reviewer #8: Yes

3. Has the statistical analysis been performed appropriately and rigorously? 

Reviewer #7: Yes

Reviewer #8: Yes

4. Have the authors made all data underlying the findings in their manuscript fully available?

Reviewer #7: Yes

Reviewer #8: Yes

5. Is the manuscript presented in an intelligible fashion and written in standard English?

Reviewer #7: Yes

Reviewer #8: Yes

6. Review Comments to the Author

Reviewer #7: 1. The authors have sufficiently addressed the feedback from the previous review round, and I believe this manuscript is now suitable for publication.

2. The manuscript presents a technically solid piece of scientific research.

3. The statistical analysis has been conducted appropriately and rigorously.

4.The authors have made all the data supporting their manuscript's findings fully accessible.

5.The manuscript is presented clearly and is written in standard English.

Reviewer #8: In the uploaded manuscript, I have included three minor comments on your Discussion section and highlighted their text in light blue so that they can be detected easily.

Also: Well done dear authors. My initial assessment was that your study brings value to the healthcare services field needs serious attention to stronger scientific writing and organization. Now, after rounds of review and your hard work, its writing matches its value. Thank you for following my numerous recommendations.

7. PLOS authors have the option to publish the peer review history of their article (what does this mean? ). If published, this will include your full peer review and any attached files.

**Do you want your identity to be public for this peer review?** For information about this choice, including consent withdrawal, please see our Privacy Policy .

Reviewer #7: No

Reviewer #8: **Yes: ** Shabnam Varmazyari, DDS-MDPH

---

## [Author Response · Author response to Decision Letter 3]

1 May 2025

Responses to the comments made by Academic Editor:

Our response: Thank you for this important observation. We have reviewed our reference list thoroughly. In our initial submission, we cited the 2000 Cochrane Review by Panpanich and Garner on growth monitoring. However, we acknowledge that this review has been withdrawn and superseded by an updated Cochrane Review.

Accordingly, we have removed the citation of the withdrawn review and have replaced it with the updated and current Cochrane Review. This change is also reflected in the revised manuscript text and reference list and read as follows (lines 10, 26, 29, 40, 83, 127, 135, 535, 583):

“1. Taylor M, Tapkigen J, Ali I, Liu Q, Long Q, Nabwera H. The impact of growth monitoring and promotion on health indicators in children under five years of age in low- and middle-income countries. Cochrane Database Syst Rev. 2023;10(10):Cd014785. Epub 20231012. doi: 10.1002/14651858.CD014785.pub2. PubMed PMID: 37823471; PubMed Central PMCID: PMCPMC10568659.”

Responses to the comments made by reviewer 8:

In the uploaded manuscript, I have included three minor comments on your Discussion section and highlighted their text in light blue so that they can be detected easily.

Also: Well done dear authors. My initial assessment was that your study brings value to the healthcare services field needs serious attention to stronger scientific writing and organization. Now, after rounds of review and your hard work, its writing matches its value. Thank you for following my numerous recommendations.

Our response: We appreciate your kind observation. We now have revised the discussion section as suggested.

Please dedicate the entire first paragraph of the discussion section to: 1) briefly restating study aims 2) briefly restating main study findings. Including one sentence on the study's importance and reserach gap is appreciated, which you have already included.

Don't include any interpretations or comparisons with the literature here, save them for the following pargaraphs.

In the following pargaraphs, each main fdining is brief restated and comapred to the literature and explained in that context, as you have already mostly done so.

So please just revise this frist paragraph to include these 3 features: 1) briefly pointing out the study's importance and reserach gap 2) briefly restating study aims 3) briefly restating main study findings

Our response: Edited as suggested, reads as follows (lines 423-430):

“This study assessed caregivers’ knowledge, perceptions, utilization, and barriers related to growth monitoring and promotion (GMP) services in rural Bangladesh. It compared GMP services delivered at home with those delivered at health facilities. The study addresses a key evidence gap by providing one of the first comparative analyses of these two service delivery platforms. Findings showed that caregivers in home-based GMP areas had greater awareness of GMP, received more GMP cards, and were more likely to have their children measured and counseled. Overall GMP service utilization was 40% in home-based areas, compared to 0% in facility-based areas. This finding underscores a stark contrast in service reach and engagement.”

This is a paragraph on your study's limitations and the ways you strived to mitigate them. Such paragraphs are typically included at the end of the discussion section. You aready have another one of these paragraphs at the end of your discussion, so I suggest you move this one to the end as well and merge the to paragraphs together. Also please make them more concise after merging them.

Our response: Edited as suggested, reads as follows (lines 517-535):

“The validity and reliability of the study findings may have been affected by several factors. Only one NGO provided home-based GMP services, and its longstanding presence in the community could have influenced caregivers’ service uptake. However, similar sociodemographic characteristics among participants have likely reduced this bias. Despite the NGO’s role and the accessibility of home-based services, utilization remained low, suggesting limited influence. We exclude children with chronic illness, severe malnutrition, and twins, which may introduce selection bias and limit generalizability. No data were collected on excluded children, so their impact on findings and GMP utilization patterns could not be assessed. Twins were excluded to minimize information bias. Though the study period was short, it still offers important insights into GMP utilization in rural settings. The study was conducted in a few rural subdistricts, excluding urban areas, which may limit the representativeness of national GMP patterns. However, two recent studies from South Asia and Africa reported no urban–rural differences in GMP utilization [25, 28]. The cross-sectional design limits causal inference between caregivers’ utilization and associated factors. The absence of longitudinal follow-up also prevents understanding of longer-term impacts. Facility-based GMP services were not utilized at all, preventing analysis of predictors in that group. Differences in baseline characteristics between groups may have contributed to outcome differences despite statistical adjustments. Thus, results should be interpreted with caution. Still, the study contributes novel evidence from a context where comparable government or research data are lacking [1].”

I belive you should move this paragraph so that it comes before the "future research"one that is currently located above it.

Our response: Edited as suggested, reads as follows (lines 536-550):

“Future research should explore these factors further to provide a more comprehensive understanding of GMP utilization dynamics. In particular, studies should explore sociocultural and behavioral influences on caregivers’ decisions—such as preferences for curative over preventive care and trust in various service providers. Further investigation is also needed into the effectiveness of health worker communication strategies, the role of incentives, and the influence of service accessibility factors like distance, cost, and competing time demands. Comparative research across urban and rural contexts would also be valuable for identifying context-specific barriers and informing the design of more tailored, equitable GMP interventions. While this study did not assess long-term outcomes, prior evidence from LMICs suggests that consistent GMP engagement—particularly when combined with responsive counseling—can lead to earlier identification of growth faltering, improved caregiver feeding practices, and ultimately better child nutrition, development outcomes and productivity [11, 28, 39]. Longitudinal research in Bangladesh is needed to explore whether improved service utilization in home-based settings translates into sustained improvements in child growth and development metrics.”

---

## [Editor Report · Decision Letter 3]

4 May 2025

Growth monitoring and promotion program services utilization patterns between home-based and facility-based delivery methods: a comparative analysis

PONE-D-24-20319R3

Dear Dr. Hossain,

We’re pleased to inform you that your manuscript has been judged scientifically suitable for publication and will be formally accepted for publication once it meets all outstanding technical requirements.

Kind regards,

Zeheng Wang

Academic Editor

PLOS ONE
---

## [Editor Report · Acceptance letter]

PONE-D-24-20319R3

PLOS ONE

Dear Dr. Hossain,

I'm pleased to inform you that your manuscript has been deemed suitable for publication in PLOS ONE. Congratulations! Your manuscript is now being handed over to our production team.

Kind regards,

on behalf of

Dr. Zeheng Wang

Academic Editor

PLOS ONE